# The impact of a health professional recommendation on weight loss attempts in overweight and obese British adults: a cross-sectional analysis

Sarah E Jackson,[1] Jane Wardle,[1] Fiona Johnson,[1] Nicholas Finer,[2] Rebecca J Beeken[1]

[1]Department of Epidemiology and Public Health, University College London, London, UK
[2]Bariatric Medicine and Surgery, University College London Institute of Cardiovascular Science, London, UK

**Correspondence to**
Dr R J Beeken;
r.beeken@ucl.ac.uk

## ABSTRACT

**Objectives:** To examine the effect that health professional (HP) advice to lose weight has on overweight and obese adults' motivation to lose weight and attempts to lose weight.

**Design:** Cross-sectional survey.

**Setting:** Great Britain.

**Participants:** 810 overweight or obese (body mass index $\geq 25$ kg/m$^2$) adults.

**Main outcome measures:** Participants were asked if they had ever received HP advice to lose weight and reported their desire to weigh less (ideal weight $\leq 95\%$ of current weight) and whether they were attempting to lose weight.

**Results:** Only 17% of overweight and 42% of obese respondents recalled ever having received HP advice to lose weight. HP advice was associated with wanting to weigh less (89% vs 61% among those not receiving advice) and attempting to lose weight (68% vs 37%). In multivariable analyses, HP advice to lose weight was associated with increased odds of wanting to weigh less (OR=3.71, 95% CI 2.10 to 6.55) and attempting to lose weight (OR=3.53, 95% CI 2.44 to 5.10) independent of demographic characteristics and weight status.

**Conclusions:** HP advice to lose weight appears to increase motivation to lose weight and weight loss behaviour, but only a minority of overweight or obese adults receive such advice. Better training for HPs in delivering brief weight counselling could offer an opportunity to improve obese patients' motivation to lose weight.

### Strengths and limitations of this study

- This is the first study to show an association between health professional advice to lose weight and weight loss attempts in a British sample.
- These results provide support for the recommendation that health professionals should discuss weight with overweight patients; highlighting the potential usefulness of this advice.
- Data were not collected on actual weight loss, so the impact of health professional advice on weight loss outcomes could not be established.

The majority of overweight and obese adults say they want to lose weight,[7] but only around half of them report actively trying to lose weight.[7 8] One potential source of motivation to lose weight is advice from a health professional. Physicians and other health professionals have a unique opportunity to raise obese patients' awareness of their weight status and its associated health risks, and to offer advice on weight management. In the USA, doctors are recommended to screen for obesity and refer all obese patients for weight loss treatment.[9] Recent reports by the Academy of Medical Royal Colleges (AMRC) and the Royal College of Physicians (RCP) have called for similar action in the UK.[10 11] Under the policy of 'making every contact count,' the AMRC recommends that health professionals should routinely speak to overweight and obese patients about diet and exercise habits at each appointment and offer help. This recommendation is also made directly to health professionals in the UK through the National Institute for Health and Care Excellence (NICE) guidelines.[12]

In spite of these recommendations, the current figures indicate that only a minority of health professionals routinely give weight

## INTRODUCTION

In most Western countries, between 40% and 70% of adults have a body weight that places them at risk of developing weight-related long-term health problems.[1 2] There is good evidence that losing as little as 5% of body weight confers significant cardiometabolic benefit for overweight and obese individuals,[3 4] although sustained weight loss is difficult to achieve.[5 6]

advice.[13–18] A range of barriers, including perceived lack of time, inadequate knowledge, lack of training or confidence and inadequate teaching materials, have been identified as contributing to the relatively low rates of weight control advice from health professionals.[19–25] Importantly, many health professionals also doubt the usefulness of providing weight advice because they do not feel it will change patient behaviour.[19 21 24 26]

However, there is evidence suggesting that health professionals can play a valuable role in helping patients to change their behaviour. The advice and involvement of health professionals in helping smokers quit is effective, and has led to evidence-based guidelines for their active intervention.[27] A recent systematic review and meta-analysis of US studies concluded that weight loss advice from a health professional in primary care is associated with positive weight loss behaviour change in overweight and obese patients,[28] but no equivalent studies have been conducted in the UK population.

This study therefore tested the hypothesis that advice to lose weight from a doctor or other health professional would be associated with a greater likelihood of trying to lose weight in a British sample of overweight and obese adults.

## METHODS
### Design and participants
Data on weight loss motivation and weight loss advice were collected from a large sample of British adults (n=1986; 932 men, 1054 women) as part of a home-based face-to-face survey from across Great Britain in April 2012. To reduce potential bias, data were collected by an independent market research company (TNS) which had no knowledge of our study aims and which asked these questions alongside questions on other topics. TNS uses a random location methodology based on the 2001 Census small-area statistics and the postcode address file, stratified by Government Office Region and social grade, to select sample points. At each location, quotas are set for age, gender, children in the home and working status to ensure a balanced sample of adults within effective contacted addresses. Interviews are carried out on weekdays between 14:00 and 20:00 and at the weekend. Interviewers are instructed to leave three doors between each successful interview.

This survey was designed as part of an ongoing study assessing changes in weight perceptions in the British population (see Johnson et al[29] for findings of the previous two surveys in this series), but it seemed timely given the recent recommendations to also use the data collected to explore the relationship between health professional advice to lose weight and weight loss motivation. The majority of respondents provided height and weight data (n=1557). These analyses focus on respondents with a body mass index (BMI) in the overweight or obese range (BMI ≥25; n=810).

## Measures
*Weight loss motivation:* Respondents were asked how much they would ideally like to weigh, with desire to weigh less indexed by reporting an ideal weight ≤95% of self-reported current weight (ie, ≥5% weight loss). They were also asked which of the following statements best described them: (1) *I'm not bothered about my weight*; (2) *I watch my weight to keep it where it is now*; (3) *I'm trying to lose weight* or (4) *I'm trying to gain weight*. Respondents indicating they were trying to lose weight were compared with the other groups combined.

*Health professional advice to lose weight:* Respondents were asked whether a doctor or other health professional had ever told them that they should lose weight (yes/no).

*Current anthropometry:* Height and weight were self-reported in metric or imperial units as the respondent preferred. BMI ($kg/m^2$) was calculated from self-reported heights and weights.

*Demographics:* Age, sex, ethnicity, marital status and socioeconomic status (SES) were included in the analyses. Owing to the small number of participants from any individual ethnic minority group, ethnicity was only categorised as white versus non-white. Marital status was categorised as married/living as married versus unmarried. SES was defined according to the National Readership Survey classification of the person's current or last occupation[30] and dichotomised into ABC1 (higher SES) versus C2DE (lower SES).

## Statistical analyses
Analyses were performed using IBM SPSS Statistics V.19, with weighted data to match the population characteristics. Variables used for weighting included age, sex, social grade and standard region. Analyses were repeated on unweighted data with no significant differences in the results, so with the exception of the sample description, only analyses on weighted data are reported.

Descriptive statistics (unweighted data) are presented for sample characteristics and prevalence of weight loss motivation and weight loss recommendations. Prevalence statistics are also reported for normal-weight (BMI 18.5–24.9) respondents in the sample (n=705) for comparison. Multivariable logistic regression (weighted data) was used to examine associations between health professional advice to lose weight and weight loss motivation controlling for age, sex, ethnicity, marital status, SES and weight status (overweight vs obese).

There was very little missing data (0.4% (n=3) for health professional advice, 9.4% (n=76) for desire to weigh less, 1.7% (n=14) for attempting to lose weight, 0% (n=0) for demographic variables). Analyses were run only for participants who had provided full data on the relevant variables (n=731, 90.2% for analyses of desire to weigh less, n=793, 97.9% for analyses on attempting to lose weight).

## RESULTS

Weight and height data were provided by 1557 of 1986 respondents, of whom 48 (3%) had a BMI within the underweight range (BMI <18.5), 699 (45%) were in the healthy weight range, 528 (34%) were overweight (BMI ≥25 and <30) and a further 282 (18%) were obese (BMI ≥30). Subsequent analyses are restricted to respondents whose BMI defined them as overweight or obese (n=810).

The demographic and anthropometric characteristics of the overweight and obese respondents are shown in table 1. Their mean age was 51.3 years (range 16–90 years); 47% were female, 89% were white, 62% were married or living as married and 61% were lower SES. Their mean weight was 85.3 kg and mean BMI was 29.6 (27.2 in overweight respondents and 34.2 in obese respondents).

A quarter (26%) of the overweight or obese respondents (17% of overweight and 42% of obese) reported ever having received a health professional's advice to lose weight. For comparison, the rate of a health professional's advice among normal-weight respondents in the sample (n=705) was 4%. Among those who were overweight or obese, mean BMI was higher among those who had received a health professional's advice to lose weight (32.1, SD 5.2) than those who had not (28.7, SD 3.5; t(276.93)=8.60, p<0.001). A health professional's advice appeared to become normative (ie, was reported by more than 50% of respondents) at a BMI of approximately 37 (see figure 1).

Overall, 68% of respondents wanted to weigh less and 45% were currently trying to lose weight. These rates are notably higher than those that were reported by normal-weight respondents, among whom 21% wanted to weigh less and 13% were trying to lose weight. Having received health professional advice to lose weight was associated with a higher prevalence of wanting to weigh less (89% vs 61%) and of attempting weight loss (68% vs 37%) in the overweight/obese sample. This pattern was the same for obese and overweight participants (see figure 2). The results of the multivariable logistic regression models are shown in table 2. Compared with overweight respondents, obese respondents were more likely to want to lose weight (OR=8.57) and also more likely to be attempting to lose weight (OR=1.91). Being female quadrupled the odds of wanting to weigh less (OR=4.39) and doubled the odds of weight loss attempts (OR=1.93). The odds of attempting weight loss were significantly lower in older respondents (OR=0.52) and higher in higher SES respondents (OR=1.40), but there were no significant associations with marital status or ethnicity. After controlling for the effects of demographics and weight status, having received a health professional's advice to lose weight more than tripled the odds of wanting to weigh less (OR=3.71) and trying to lose weight (OR=3.53).

**Table 1** Demographic and anthropometric characteristics of a sample of overweight and obese British adults (n=810)

| Demographic and healthcare characteristics | |
| --- | --- |
| Age (years) | |
| Mean (SD) | 51.3 (17.9) |
| <55 | 447 (55.2) |
| ≥55 | 363 (44.8) |
| Sex | |
| Male | 432 (53.3) |
| Female | 378 (46.7) |
| Ethnicity | |
| White | 721 (89.0) |
| Non-white | 88 (10.9) |
| Marital status | |
| Unmarried | 312 (38.5) |
| Married/living as married | 498 (61.5) |
| SES | |
| Lower | 493 (60.9) |
| Higher | 317 (39.1) |
| Health professional advice to lose weight | |
| No | 599 (74.0) |
| Yes | 208 (25.7) |
| **Anthropometric characteristics** | |
| Mean (SD) height (cm) | 169.4 (10.7) |
| Mean (SD) weight (kg) | 85.3 (16.2) |
| Mean (SD) body mass index (kg/m$^2$) | 29.6 (4.3) |
| Weight status | |
| Overweight | 528 (65.2) |
| Obese | 282 (34.8) |

Values are numbers (percentages) unless stated otherwise unweighted data shown.
Numbers may not sum to the total sample number, as some items were not answered by all participants.
Percentages were derived from the full sample and may therefore not sum to 100%.
SES, socioeconomic status.

## DISCUSSION

In this population-based sample of overweight and obese British adults, only a quarter (26%) reported having received advice to lose weight from a health professional. Around two-thirds reported a desire to weigh at least 5% less than their current weight, and just under half said they were actively trying to lose weight. Having received health professional advice to lose weight was strongly associated with wanting to weigh less and trying to lose weight after controlling for demographic characteristics.

Healthcare providers in primary care in the UK are required to record BMI on all patients and also recommended to discuss diet and exercise with all overweight and obese patients.[12] However, in this sample, fewer than half (42%) of the obese respondents and only 17% of the overweight respondents reported having been recommended by a doctor or other health professional to lose weight; these findings are similar to US studies.[13–18] Nonetheless, this is considerably higher than the 5% (among overweight respondents) and 16% (among

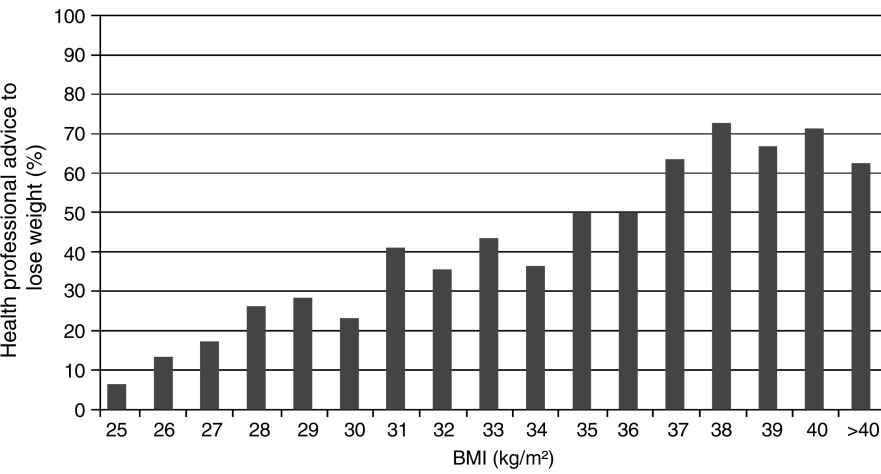

**Figure 1** Percentage of overweight and obese British adults reporting ever having received health professional advice to lose weight by body mass index. Unweighted data shown. BMI points were rounded down such that a BMI point of 25 includes 25.0–25.9.

obese respondents) reported in a similar British survey in 2001.[8] Health professional advice to lose weight became normative (ie, observed in over 50% of respondents) at a BMI of 37. This is consistent with previous research showing that health professionals often do not pay attention to weight until a patient is at a much higher BMI than what is actually defined as obese.[31–33] Health professionals report an array of barriers to providing weight loss advice, including perceived lack of time, knowledge, training and confidence.[19–25] Perhaps most importantly, obesity treatment is often perceived by physicians to be a daunting or even futile task,[26] and many say they find it professionally unrewarding.[34–38] This could well affect their enthusiasm to broach the topic.

Despite the notorious difficulty in achieving and maintaining significant weight loss, one study found that

almost half of the overweight and obese individuals believed they could lose weight if they felt they needed to.[21] This may highlight an important role for health professionals to give clear advice to patients when weight loss is needed. Our results demonstrate that receiving health professional advice to lose weight was associated with desiring a lower body weight and, more importantly, with attempting weight loss. ORs were similar to those observed in a meta-analysis of US studies.[28] Among respondents who desired to weigh less, health professional advice was associated with being significantly more likely to be trying to lose weight. The translation of behavioural intentions to changes in behaviour is known to be a major block in lifestyle interventions.[39] The finding observed here suggests that in the case of weight loss, advice from a health professional can help to bridge the intention–behaviour gap.

Together, these results provide strong support for the recommendation that physicians and other health professionals should discuss weight with overweight and obese patients.[9 10] Targeted education and training programmes on weight counselling for health professionals could help overcome some of the barriers that hold them back from 'making every contact count'.[10]

The findings of this study are subject to several limitations. We do not have response rate information because of the method of sampling, and it is not possible to know whether those who declined to participate differed from those who agreed to take part. However, as the primary focus is the association between weight loss advice and weight loss attempts, the sampling method is not likely to influence the findings. However, in terms of prevalence of weight loss advice and weight loss attempts, the results only reflect the experience of individuals who agreed to take part, which may overestimate levels of weight concern. The use of self-reported weights and heights means that BMI is most likely to be underestimated.[40 41] Consistent with this, the prevalence of obesity was lower than the figures based on measured anthropometric data from the most recent Health Survey for England[42] (18% in the present study vs 26%

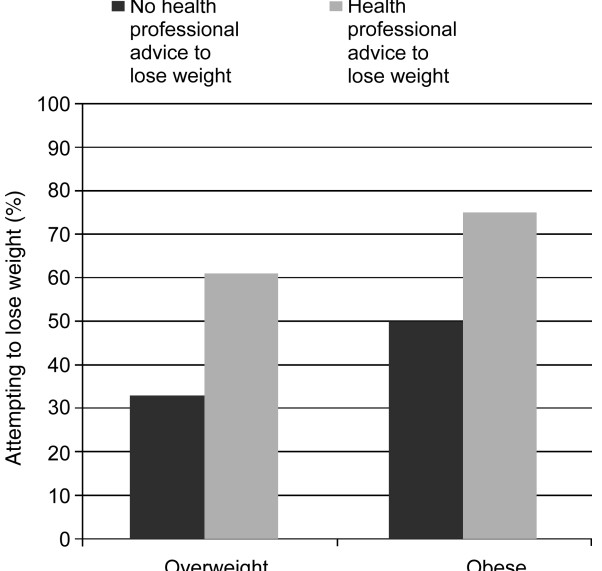

**Figure 2** Prevalence of attempting to lose weight in overweight and obese British adults by health professional advice to lose weight (yes/no) and weight status (overweight/obese).

**Table 2** Multivariable logistic regression models predicting a desire to weigh less and attempting to lose weight in a sample of overweight and obese British adults (n=810)

| Characteristics | Desire to weigh less | | | Attempting to lose weight | | |
|---|---|---|---|---|---|---|
| | OR | 95% CI | p value | OR | 95% CI | p value |
| Age (years) | | | | | | |
| <55 | 1.00 | – | – | 1.00 | – | – |
| ≥55 | 0.84 | 0.57 to 1.23 | 0.369 | 0.52 | 0.38 to 0.71 | <0.001 |
| Sex | | | | | | |
| Male | 1.00 | – | – | 1.00 | – | – |
| Female | 4.39 | 2.94 to 6.56 | <0.001 | 1.93 | 1.43 to 2.63 | <0.001 |
| Ethnicity | | | | | | |
| White | 1.00 | – | – | 1.00 | – | – |
| Non-white | 0.96 | 0.51 to 1.84 | 0.912 | 1.11 | 0.67 to 1.85 | 0.686 |
| Marital status | | | | | | |
| Unmarried | 1.00 | – | – | 1.00 | – | – |
| Married/living as married | 0.87 | 0.59 to 1.29 | 0.483 | 1.34 | 0.98 to 1.84 | 0.065 |
| SES | | | | | | |
| Lower | 1.00 | – | – | 1.00 | – | – |
| Higher | 1.34 | 0.92 to 1.95 | 0.125 | 1.40 | 1.03 to 1.89 | 0.030 |
| Weight status | | | | | | |
| Overweight | 1.00 | – | – | 1.00 | – | – |
| Obese | 8.57 | 4.87 to 15.08 | <0.001 | 1.91 | 1.38 to 2.64 | <0.001 |
| Health professional advice to lose weight | | | | | | |
| No | 1.00 | – | – | 1.00 | – | – |
| Yes | 3.71 | 2.10 to 6.55 | <0.001 | 3.53 | 2.44 to 5.10 | <0.001 |

Data were weighted on age, sex, social grade and standard region. Results were not notably different when analyses were run on unweighted data.
All variables entered into the models are shown in the table; there were no additional covariates.
SES, socioeconomic status.

in the HSE). This may limit the extent to which findings can be generalised to the entire overweight and obese population, and results should therefore be interpreted as potentially applying to a population with a higher BMI but by an unknown amount.

The finding that only a minority of overweight and obese people had been advised to lose weight was important, but it is possible that more had received advice than recalled it. Health professional-reported rates of weight counselling are markedly higher than the 40–50% reported by obese patients,[43] although there is some evidence to suggest that health professionals may overestimate the level of intervention they provide to patients.[43] Guiding health professionals on how to provide weight control advice in a way that resonates more strongly with their patients could improve the effectiveness of the advice and reduce any reservations about its utility.

These data do not address the reasons for being given or not being given weight loss advice. People who had not received advice may not have seen a health professional since they became overweight or obese; they may have seen a health professional who did not identify their weight status; or the health professional may have recognised their overweight/obesity but had reasons not to mention it. In addition, we had no data on how recently participants had seen a health professional or, for those who did report receiving advice, on the type of health

professional it had come from (general practitioner (GP), hospital physician, nurse, dietitian, pharmacist, etc). In Britain, patients typically visit their GP on registering, and subsequently only when seeking help for a specific health concern (unless they require regular check-ups for a long-term medical condition). Patients who have never registered with a primary care physician (rather have only been to specialists or A&E), or patients who have not seen one in a long time, may not have had the opportunity to have weight loss addressed. Advice may be more or less effective coming from a certain type of health professional, but it was not possible to explore this using the data we had available. Identifying the determinants of giving weight loss advice could further help tailor training for health professionals.

There was also no information on how recently participants had received advice from a health professional to lose weight. It is unclear whether 'ever' having received advice is important, or whether the timing of advice within the last week/month/year/decade was important. This is an avenue for exploration in future research.

The use of a cross-sectional design meant we were not able to determine whether health professional advice increased motivation to lose weight on an individual level. Individuals who are already concerned about their health may visit their doctors more often, and therefore have more opportunities to elicit weight loss advice, and they may even have asked for advice directly. The design

also precluded the assessment of weight loss success following health professional advice. Before the present results can be taken forward to guide policy, there needs to be evidence that advice from a health professional also leads to successful weight loss outcomes, and exploration of how the nature, extent and cost of that advice, and from whom, relates to the extent and frequency of that success. Advice-only interventions have been shown to be less effective than more intensive interventions in producing positive weight loss outcomes,[5] but previous research has demonstrated better long-term weight outcomes in treatment programmes following a medical trigger for weight loss,[28] and extrapolation from studies of GP advice for smoking cessation[27] gives some cause for optimism. Prospective longitudinal research is needed to provide insight into motivational changes and actual weight reduction following advice to lose weight.

The results of this study confirm that many overweight and obese adults in Great Britain express a desire to weigh less than they do, but notably fewer are actively trying to lose weight. Advice from a health professional was strongly associated with attempting weight loss, supporting the recommendation that health professionals should discuss weight with overweight and obese patients as a matter of routine. Better training for health professionals in discussing weight issues could make a significant contribution to population weight management.

**Contributors** Everyone listed as an author fulfils all three of the ICMJE guidelines for authorship: (1) substantial contributions to the conception and design, acquisition of data, or analysis and interpretation of data; (2) drafting of the article or revising it critically for important intellectual content; and (3) final approval of the version to be published. SEJ, JW, FJ, NF and RJB were responsible for the study concept and design. JW obtained the funding. JW and FJ acquired the data. SEJ, JW, FJ, NF and RJB analysed and interpreted the data. SEJ performed the statistical analysis. SEJ drafted the manuscript, and all authors revised it for important intellectual content. All the authors had final approval of the version to be published. JW is the guarantor. All the authors had full access to all of the data (including statistical reports and tables) in the study and can take responsibility for the integrity of the data and the accuracy of the data analysis.

**Funding** This study was supported by grants from the UK Medical Research Council, UK Economic and Social Research Council, and Cancer Research UK.

**Competing interests** None.

**Provenance and peer review** Not commissioned; externally peer reviewed.

**Data sharing statement** Full dataset available from the corresponding author at r.beeken@ucl.ac.uk.

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
