## [Reviewer comments · BMJ Open]

Some articles will have been accepted based in part or entirely on reviews undertaken for other BMJ Group journals. These will be reproduced where possible.

ARTICLE DETAILS

TITLE (PROVISIONAL)	The impact of a health professional recommendation on weight loss attempts in overweight and obese British adults: a cross-sectional analysis
AUTHORS	Jackson, Sarah; Wardle, Jane; Johnson, Fiona; Finer, Nicholas; Beeken, Rebecca

VERSION 1 - REVIEW

REVIEWER	Stephanie A. Rose, MD, MPH University of Kentucky, USA No competing Interests. However, I completed a similar study here and am actually cited in the paper.
REVIEW RETURNED	18-Sep-2013

GENERAL COMMENTS	The purpose of this study was to examine the likelihood of weight loss attempt in a population of UK overweight and obese adults based on receipt of weight loss advice from their health care provider. I think this is an important topic, because providers still feel that they do not make a difference when they discuss weight loss with their patients, despite studies suggesting otherwise. While this topic has been addressed in previous studies, according to the authors it has not been examined in a British population. The authors results from a home-based, face to face survey from across Great Britain. The subjects were asked whether a doctor had ever told them that they should lose weight. They also asked the subjects how they felt about their current weight (i.e., whether they were concerned about their weight and whether they wanted to change it). Unfortunately, despite the fact that this was a face to face interview, weights and heights were self-report. The authors found that a little over half of the patients were overweight and obese. Of these patients, they found that only a quarter of the patients overall, and only 17% of overweight patients, were told by their doctor that they needed to lose weight. Interestingly, the authors presented the BMI value at which the advice appeared to become normative, which was around a BMI of 37 kg/m². This is reminiscent of a previous study which suggested that health care providers underestimate patient weight and do not pay attention to weight until a patient is at a much higher BMI level than what is actually defined as obese. The authors should try to look for this and could discuss this in their strengths. The authors' odds ratios were similar to previous studies. The authors looked at the results both weighted and non weighted and
---

	received similar results. The figures and tables are helpful; however, in order to be able to read them independently of the manuscript, they should have complete keys at the bottom or top of the figures or tables listing all the items the authors used to weight the data or control for in the analysis in that particular table. One weakness of this study is that there is no mention of whether these subjects were asked if they had seen a primary care physician within the past 6 months or a year. Patients who have never seen a primary care physician (rather have only been to specialists or have only been to the ER), or patients who have not seen one in a long time, may not have had the opportunity to have weight loss addressed. Perhaps this was asked, and if so, it should be addressed and perhaps patients who have never seen a PCP or not within the past 6 months or a year should be taking out of the study – I am not stating that this has to be done but this question should be addressed at least in the discussion. Furthermore, I am not familiar with the medical system in Great Britain, and do not know if patients are typically seen by general practitioners for their routine care or a regular basis. Perhaps this could be briefly addressed in the introduction or in the discussion for those of us not familiar. Overall, I think this is a well-written, clear and concise paper that requires only minor revisions.
--	--

REVIEWER	John Reckless Consultant Endocrinologist Hon Reader in Medicine, University of Bath Wolfson Centre, Royal United Hospital, Bath
REVIEW RETURNED	19-Sep-2013

GENERAL COMMENTS	Comments:  1. The authors assess whether ever having received prior advice to lose weight from a health professional affected the desire to lose weight or to attempt to lose weight. 2. Participants were derived from 1,986 individuals who were surveyed at home face-to-face by an independent market research company in a separate study to examine changes in public perceptions of overweight, compared to similar surveys in 1999 and 2007. 3. Although this was surveying views about overweight only 78% (1,557) provided height and weight data. 4. Of these 1,557 this paper relates to the 52% (810) of respondents (41% of the original population) who were overweight or obese (BMI ≥ 25 kg/m²). 5. The survey was set up to "...ensure a balanced sample of adults..." from quotas for age, gender and children in the home. It is unclear the time of day when the surveys were carried out and whether on weekdays or weekends. Personally well aware of how often houses that are called upon have no individual at home, I would like to know how representative the survey can be of the
--

general population. Employment may be impaired for the severely obese. The interviewers had instructions to leave three doors after a successful interview, indicating also therefore the issue of “failed interviews”.

6. These were self-reported weights and heights, and it is very well recognised that weight is under-reported and height is over-estimated. The authors recognise this, but it remains an issue in the interpretation of the data. While self-reported weights in a population have increased significantly (Ref 31) so also has the weight at which individuals consider themselves overweight. Furthermore, despite this increase in weight, over time fewer people have recognised themselves as overweight.

7. This report therefore considers self-reported overweight and obese values, and results will need to be interpreted as applying to a population with a significantly higher BMI but by an unknown amount.

8. The authors maintain that the study respondents had “...no knowledge of our study aims...”. This may be narrowly true, but the survey was clearly related to the respondents’ weights by the nature of the questions.

9. Respondents were asked whether they had ever had health professional advice to lose weight. It is unclear whether “ever” is important, or whether timing of the advice in the last week/month/year/decade was important.

10. It is not reported as to whether the type of health professional mattered – whether GP/hospital doctor/nurse/HCA/dietitian/pharmacist/other had more or less effect.

11. Socioeconomic status was recorded as stated but it is not stated as to whether the respondents found by the interviewers closely matched the 2001 Census small area expected statistics.

12. Analyses were run only for those with full data. The biggest area of missing data seems to be “desire to weigh less”. The total number/percentage of missing individuals should be stated.

13. It would be of interest, as a control group, to have the data recorded on those who were of normal weight (BMI 18.5-24.9 kg/m²) as to “advised to lose weight”, “desire to lose weight” and “attempting to lose weight”. A proportion of these normal weight individuals will be positive to these questions, and some of them will have been over-optimistic as to their self-reported BMI.

14. The actual number of individuals who reported receiving advice was quite small being 90 overweight (17% of 528) and 118 obese (42% of 282) individuals.

15. Despite these quite small numbers obese were more likely to wish to lose weight and to be trying to lose weight than the overweight (but without control data for the self-reported “normal weight) individuals, and perhaps as expected higher in females and those at high socioeconomic status.

16. Health care professional advice in these fairly small numbers was associated with a three-fold increase in both desire and attempt.

	17. The authors discuss the biases and limitations of their study. 18. The authors state that "...Healthcare providers in primary care are required to record BMI ...recommended to discuss diet and exercise...". This may be so, but the survey in this paper was carried out in April 2012. The NICE guidelines were 2006 (ref 12), but the SIGN guidelines (ref 14) were 2010 and funding through QOF (ref 13) was not until 2013. 19. The main problem with this study is that this is a self-reported cross-sectional survey. There is no data as to whether the "desire" and the "attempt" were associated with any "success" in losing weight. Before one can take this data forward to guide policy there would need to be such evidence. Furthermore, one would need to relate the nature, extent and cost of that advice, and from whom, to the extent and frequency of that "success". 20. On page 10 (lines 17-20; ref 23) the authors note the "notorious difficulty" in achieving/maintaining weight loss, but of course this study in no way addresses this. 21. We are left with the authors' wish to believe that simple prior advice at any previous time point will lead to a successful outcome. The final paragraph of the discussion "...supporting the recommendation that health professionals should discuss weight...as a matter of routine. I would not disagree with this principle, but the authors will know (such as NF) how intensive the interventions needs to be to show positive outcomes. 22. Very minor issue – table 1 – we should not have 100.2% for male/female 23. Very minor issue – authors names/initials – As only first & last name given in author list then in contributor statement it should be "SY" and "RB".
--	--

VERSION 1 – AUTHOR RESPONSE

Reviewer: Stephanie A. Rose, MD, MPH
University of Kentucky, USA

The purpose of this study was to examine the likelihood of weight loss attempt in a population of UK overweight and obese adults based on receipt of weight loss advice from their health care provider. I think this is an important topic, because providers still feel that they do not make a difference when they discuss weight loss with their patients, despite studies suggesting otherwise. While this topic has been addressed in previous studies, according to the authors it has not been examined in a British population. The authors results from a home-based, face to face survey from across Great Britain. The subjects were asked whether a doctor had ever told them that they should lose weight. They also asked the subjects how they felt about their current weight (i.e., whether they were concerned about their weight and whether they wanted to change it). Unfortunately, despite the fact that this was a face to face interview, weights and heights were self-report.

The authors found that a little over half of the patients were overweight and obese. Of these patients, they found that only a quarter of the patients overall, and only 17% of overweight patients, were told

by their doctor that they needed to lose weight. Interestingly, the authors presented the BMI value at which the advice appeared to become normative, which was around a BMI of 37 kg/m². This is reminiscent of a previous study which suggested that health care providers underestimate patient weight and do not pay attention to weight until a patient is at a much higher BMI level than what is actually defined as obese. The authors should try to look for this and could discuss this in their strengths.

Response: We also feel this is an interesting finding that warrants attention in the discussion, so we have added the following (paragraph 2, page 8): “Health professional advice to lose weight became normative (i.e. observed in over 50% of respondents) at a BMI of 37. This is consistent with previous research showing that health professionals often do not pay attention to weight until a patient is at a much higher BMI than what is actually defined as obese.[31–33]”

The authors’ odds ratios were similar to previous studies. The authors looked at the results both weighted and non weighted and received similar results. The figures and tables are helpful; however, in order to be able to read them independently of the manuscript, they should have complete keys at the bottom or top of the figures or tables listing all the items the authors used to weight the data or control for in the analysis in that particular table.

Response: Thank you - we have added more information below each table and figure (pages 11 to 14), and hope this helps in their interpretation when viewed independently of the manuscript.

One weakness of this study is that there is no mention of whether these subjects were asked if they had seen a primary care physician within the past 6 months or a year. Patients who have never seen a primary care physician (rather have only been to specialists or have only been to the ER), or patients who have not seen one in a long time, may not have had the opportunity to have weight loss addressed. Perhaps this was asked, and if so, it should be addressed and perhaps patients who have never seen a PCP or not within the past 6 months or a year should be taking out of the study – I am not stating that this has to be done but this question should be addressed at least in the discussion.

Response: This is an interesting point. Unfortunately, we did not have data on how recently participants had seen a primary care physician so were unable to run analyses excluding those individuals who had not recently seen one. However, we have now raised this issue in the discussion (paragraph 7, page 9), stating: “In addition, we had no data on how recently participants had seen a health professional ... Patients who have never registered with a primary care physician (rather have only been to specialists or A&E), or patients who have not seen one in a long time, may not have had the opportunity to have weight loss addressed.”

Furthermore, I am not familiar with the medical system in Great Britain, and do not know if patients are typically seen by general practitioners for their routine care or a regular basis. Perhaps this could be briefly addressed in the introduction or in the discussion for those of us not familiar.

Response: Thank you for highlighting this. We have added the following into the discussion (paragraph 7, page 9) to inform readers who are not familiar with the British medical system: “In Britain, patients typically visit their GP upon registering, and subsequently only when seeking help for a health concern (unless they require regular check-ups for a long-term medical condition).”

Overall, I think this is a well-written, clear and concise paper that requires only minor revisions.

Reviewer: John Reckless
Consultant Endocrinologist
Hon Reader in Medicine, University of Bath Wolfson Centre, Royal United Hospital, Bath

Comments:

1. The authors assess whether ever having received prior advice to lose weight from a health professional affected the desire to lose weight or to attempt to lose weight.
2. Participants were derived from 1,986 individuals who were surveyed at home face-to-face by an independent market research company in a separate study to examine changes in public perceptions of overweight, compared to similar surveys in 1999 and 2007.
3. Although this was surveying views about overweight only 78% (1,557) provided height and weight data.
4. Of these 1,557 this paper relates to the 52% (810) of respondents (41% of the original population) who were overweight or obese (BMI ≥ 25 kg/m²).
5. The survey was set up to "...ensure a balanced sample of adults..." from quotas for age, gender and children in the home. It is unclear the time of day when the surveys were carried out and whether on weekdays or weekends. Personally well aware of how often houses that are called upon have no individual at home, I would like to know how representative the survey can be of the general population. Employment may be impaired for the severely obese. The interviewers had instructions to leave three doors after a successful interview, indicating also therefore the issue of "failed interviews".

Response: As requested, we have added information on the time of day the surveys were carried out, and whether on weekdays or weekends: "Interviews are carried out on weekdays between 2pm and 8pm and at the weekend" (method, paragraph 1, page 5).

However, we have no information on the number of attempted interviews that we can compare to the number that were successful, and it is not possible to know whether those who refused to participate differed on any socio-demographic variable from those who agreed to take part. We agree that this is a limitation of this method of data collection, and have acknowledged this in the discussion: "We do not have response rate information because of the method of sampling, and it is not possible to know whether those who declined to participate differed from those who agreed to take part" (paragraph 5, page 9).

There did not appear to be an overrepresentation of obese respondents in the sample, as might be expected if these people are more likely to be at home and therefore available for interview. In fact, there was a lower rate of obesity in the sample than might be expected, according to a recent estimate by the Health Survey for England (see discussion, paragraph 5, page 9), although we were reliant on self-reported measures of anthropometry which are likely to underestimate the true rate of obesity in the sample.

6. These were self-reported weights and heights, and it is very well recognised that weight is under-

reported and height is over-estimated. The authors recognise this, but it remains an issue in the interpretation of the data. While self-reported weights in a population have increased significantly (Ref 31) so also has the weight at which individuals consider themselves overweight. Furthermore, despite this increase in weight, over time fewer people have recognised themselves as overweight.

Response: We agree this is an important issue. By calculating weight status from respondents' self-reported height and weight (rather than asking them to report their own weight status) we should have overcome, at least to some extent, the problem relating to weight misperception. However, it is still important to consider that weight was likely to have been underreported across the sample, and as a result BMI and the proportion with overweight/obesity will have been underestimated. We make this point in the discussion: "Use of self-reported weights and heights mean that BMI is likely to be underestimated.[40,41] ... This may limit the extent to which findings can be generalised to the entire overweight and obese population..." (paragraph 5, page 9).

7. This report therefore considers self-reported overweight and obese values, and results will need to be interpreted as applying to a population with a significantly higher BMI but by an unknown amount.

Response: Thank you for this comment. We had acknowledged the limitations of self-reported data in the discussion, but feel that adding "... results should therefore be interpreted as potentially applying to a population with a higher BMI but by an unknown amount," as suggested, more clearly describes the implications of this limitation (paragraph 4, page 9).

8. The authors maintain that the study respondents had "...no knowledge of our study aims...". This may be narrowly true, but the survey was clearly related to the respondents' weights by the nature of the questions.

Response: We apologise for not having been clearer about the data collection. We have edited the method section (paragraph 1, page 5) to explain that the questions on weight and weight perceptions were asked among other questions from other researchers not related to this study: "To reduce potential bias, data were collected by an independent market research company (TNS) that had no knowledge of our study aims and who asked these questions alongside questions on other topics."

9. Respondents were asked whether they had ever had health professional advice to lose weight. It is unclear whether "ever" is important, or whether timing of the advice in the last week/month/year/decade was important.

Response: We did not have data available to answer this question, but have raised the issue in the discussion (paragraph 8, page 10), commenting: "There was also no information on how recently participants had received advice from a health professional to lose weight. It is unclear whether "ever" having received advice is important, or whether the timing of advice within the last week/month/year/decade was important. This is an avenue for exploration in future research."

10. It is not reported as to whether the type of health professional mattered – whether GP/hospital doctor/nurse/HCA/dietitian/pharmacist/other had more or less effect.

Response: This is a very interesting comment. Unfortunately we do not have data available on the source of health professional advice so are unable to explore this in this sample, but we have added a comment in the discussion (paragraph 7, page 9/10) raising this issue: "In addition, we had no data on

... the type of health professional it had come from (GP, hospital physician, nurse, dietitian, pharmacist, etc.). ... Advice may be more or less effective coming from a certain type of health professional, but it was not possible to explore this using the data we had available.”

11. Socioeconomic status was recorded as stated but it is not stated as to whether the respondents found by the interviewers closely matched the 2001 Census small area expected statistics.

Response: We thank the reviewer for this suggestion. This is a good idea but given our results have been analysed on data weighted to match the general population, we are not sure this would add to the article. In addition, because SES was determined according to the National Readership Survey classification in the present study, and by the National Statistics Socio-economic Classification (NS-SEC) in the UK Census, it is difficult to compare the two. If the reviewer feels it is important to include nonetheless, then we will happily do so.

12. Analyses were run only for those with full data. The biggest area of missing data seems to be “desire to weigh less”. The total number/percentage of missing individuals should be stated.

Response: We have added the number of missing individuals on each variable, and the total number of participants who had full data for analyses of desire to weigh less and attempting to lose weight, into the method (last paragraph, page 6).

13. It would be of interest, as a control group, to have the data recorded on those who were of normal weight (BMI 18.5-24.9 kg/m²) as to “advised to lose weight”, “desire to lose weight” and “attempting to lose weight”. A proportion of these normal weight individuals will be positive to these questions, and some of them will have been over-optimistic as to their self-reported BMI.

Response: As requested, we have added statistics into the results section on the proportion of normal weight respondents who had been advised to lose weight (paragraph 3, page 7) and the proportion who wanted to weigh less and were attempting to lose weight (paragraph 4, page 7).

14. The actual number of individuals who reported receiving advice was quite small being 90 overweight (17% of 528) and 118 obese (42% of 282) individuals.

15. Despite these quite small numbers obese were more likely to wish to lose weight and to be trying to lose weight than the overweight (but without control data for the self-reported “normal weight) individuals, and perhaps as expected higher in females and those at high socioeconomic status.

16. Health care professional advice in these fairly small numbers was associated with a three-fold increase in both desire and attempt.

17. The authors discuss the biases and limitations of their study.

18. The authors state that “...Healthcare providers in primary care are required to record BMI ...recommended to discuss diet and exercise...”. This may be so, but the survey in this paper was

carried out in April 2012. The NICE guidelines were 2006 (ref 12), but the SIGN guidelines (ref 14) were 2010 and funding through QOF (ref 13) was not until 2013.

Response: Given the reviewer's concerns about the latter two references, and whether health professionals would have been aware of these guidelines at the time survey data were collected, we have removed these references from the manuscript, leaving only the reference for NICE guidelines.

19. The main problem with this study is that this is a self-reported cross-sectional survey. There is no data as to whether the "desire" and the "attempt" were associated with any "success" in losing weight. Before one can take this data forward to guide policy there would need to be such evidence. Furthermore, one would need to relate the nature, extent and cost of that advice, and from whom, to the extent and frequency of that "success".

Response: We feel this is a very good point, and have added the following into the discussion (paragraph 9, page 10): "Before the present results can be taken forward to guide policy, there needs to be evidence that advice from a health professional also leads to successful weight loss outcomes; and exploration of how the nature, extent, and cost of that advice, and from whom, relates to the extent and frequency of that success."

20. On page 10 (lines 17-20; ref 23) the authors note the "notorious difficulty" in achieving/maintaining weight loss, but of course this study in no way addresses this.

Response: Unfortunately, it was not possible to address this issue using the data available. However, we have mentioned the need to explore whether weight loss is achieved in the discussion (paragraph 9, page 10).

21. We are left with the authors' wish to believe that simple prior advice at any previous time point will lead to a successful outcome. The final paragraph of the discussion "...supporting the recommendation that health professionals should discuss weight...as a matter of routine. I would not disagree with this principle, but the authors will know (such as NF) how intensive the interventions needs to be to show positive outcomes.

Response: We do acknowledge the limited success of most weight loss interventions, and agree that more intensive intervention would likely be required to induce positive weight loss outcomes, so we have added the following to the previous paragraph: "Advice-only interventions have been shown to be less effective than more intensive interventions in producing positive weight loss outcomes [46], ..." (paragraph 9, page 10).

22. Very minor issue – table 1 – we should not have 100.2% for male/female

Response: Thank you for bringing this typo to our attention! The percentage of male respondents has now been corrected from 53.5% to 53.3% (Table 1, page 11).

23. Very minor issue – authors names/initials – As only first & last name given in author list then in contributor statement it should be "SY" and "RB".

Response: Thank you for pointing this out.